# Hesperetin Inhibits Porcine Reproductive and Respiratory Syndrome Virus Replication by Downregulating the P38/JUN/FOS Pathway In Vitro

**DOI:** 10.3390/microorganisms13020450

**Published:** 2025-02-18

**Authors:** Ruiheng Gu, Feike Zhao, Quanying Li, Limin Hou, Guochang Liu, Xueyan Sun, Junyuan Du, Binghu Fang

**Affiliations:** College of Veterinary Medicine, South China Agricultural University, Guangzhou 510642, China; guruiheng@stu.scau.edu.cn (R.G.); zfk160099@163.com (F.Z.); 185694523l@163.com (Q.L.); 19120536126@163.com (L.H.); liuguoc@stu.scau.edu.cn (G.L.); xueyansun@163.com (X.S.); dujunyuan@stu.scau.edu.cn (J.D.)

**Keywords:** Hesperetin (Hst), Porcine Reproductive and Respiratory Syndrome Virus (PRRSV), inhibits, P38 MAPK pathway, P38/JUN/FOS

## Abstract

Porcine reproductive and respiratory syndrome virus (PRRSV) is a persistent pathogen that causes significant economic losses to the global swine industry. Commercial vaccines provide only partial protection, and no effective therapeutic treatments are currently available. In this study, we demonstrate that hesperetin (Hst), a flavonoid glycoside derived from orange and tangerine peels, inhibits PRRSV replication in a dose-dependent manner in Marc-145 and PAMs cells. Transcriptome analysis further reveals that the anti-PRRSV effects of Hst are associated with the suppression of the P38 MAPK pathway, as Hst significantly downregulates key genes, including NRA41, JUN, FOS, and DUSP1. Subsequent investigations show that Hst inhibits PRRSV replication by downregulating the P38/JUN/FOS signaling cascade. These findings offer valuable insights for the development of novel preventive and therapeutic strategies against PRRSV infection.

## 1. Introduction

Porcine reproductive and respiratory syndrome (PRRS) is a viral disease caused by the Porcine reproductive and respiratory syndrome virus (PRRSV), which poses a significant threat to the global swine industry [1]. PRRSV causes reproductive disorders in sows and a range of respiratory symptoms in pigs of all ages, with severe cases leading to death. The mortality rate can range from 2% to 100% [2]. PRRSV belongs to the order *Nidovirales*, the family *Arteriviridae*, and the genus *Arterivirus*. It is a single-stranded, positive-sense RNA virus [3]. The PRRSV genome consists of a polycistronic, positive-sense RNA approximately 14.9–15.5 kb in length and contains 10 overlapping open reading frames (ORFs): ORF1a, ORF1b, ORF2, ORF2b, ORF3, ORF4, ORF5, ORF5a, ORF6, and ORF7. The 5′ and 3′ ends of the genome contain untranslated regions (UTRs), with a 5′ cap structure and a 3′ poly-A tail. The genome encodes 14 non-structural proteins and 8 structural proteins [1,4]. PRRSV currently has a single serotype, which can be further classified into two genotypes—European type (EU-type) and North American type (NA-type)—based on genetic evolution, with approximately 60% nucleotide similarity between the two genotypes [5,6]. Due to the significant economic losses caused by the NA-type PRRSV in China, it has been the focus of extensive research. For instance, the highly pathogenic PRRSV outbreak in 2006 led to the infection of 2 million pigs and the death of 400,000 pigs in China [7].

PRRSV is highly prone to mutations, resulting in significant genetic diversity. Currently, at least 9 lineages of NA-type PRRSV and 4 lineages of EU-type PRRSV exist worldwide, presenting a considerable challenge for PRRSV control [8]. Vaccination is a commonly used strategy to protect pig herds from PRRSV, with inactivated and modified live vaccines being the most widely applied [9,10]. However, currently commercialized inactivated PRRSV vaccines fail to induce robust protective immunity in pigs, primarily due to factors such as low levels of neutralizing antibodies, delayed antibody production, and weak cell-mediated immune responses [11,12]. Although modified live vaccines overcome some of the limitations of inactivated vaccines, they still present challenges, including incomplete protection against heterologous wild-type strains, the risk of vertical and horizontal transmission of the vaccine virus, reversion to virulence, and potential recombination between the vaccine virus and wild-type strains [13]. Therefore, PRRSV remains a significant challenge for the swine industry, highlighting the urgent need for new strategies to prevent PRRSV infection.

The MAPK pathway is evolutionarily conserved in both plants and animals, playing a crucial role in various cellular processes, including proliferation, differentiation, apoptosis, angiogenesis, and tumor metastasis. It is also integral to activating innate immunity against diverse pathogens [14]. However, emerging evidence indicates that during viral infections, activation of the MAPK pathway can facilitate viral replication. For instance, viruses such as severe fever with thrombocytopenia syndrome virus (SFTSV), influenza virus (IV), herpes simplex virus type 1 (HSV-1), dengue virus (DENV), and severe acute respiratory syndrome coronavirus 2 (SARS-CoV-2) have been shown to activate this pathway [15,16,17]. Four branches of the MAPK signaling pathway have been identified: ERK, JNK, P38 MAPK, and ERK5 [18]. As an immune system disorder, PRRSV also activates the MAPK pathway [19,20]. Inhibition of the MAPK pathway has been shown to suppress PRRSV infection, significantly reducing viral genomic and subgenomic RNA synthesis, as well as viral protein production and progeny virus release [21,22]. Recently, Hu et al. demonstrated that allicin treatment can reverse the upregulation of pro-inflammatory, TNF, and MAPK signaling pathways induced by PRRSV infection, enhancing anti-PRRSV activity [23]. These findings suggest that inhibiting the MAPK pathway may serve as an effective strategy to combat PRRSV infection.

Numerous natural compounds, including xanthohumol, matrine, quercetin, artemisinin, and epicatechin, have been shown to exhibit anti-PRRSV activity [24,25,26,27,28]. However, there are currently no commercially available drugs to prevent PRRSV infection. Consequently, there is an urgent need to develop stable, universally effective drug-based prevention strategies against PRRSV. Hesperetin (Hst), a flavonoid glycoside, phytoestrogen, and bioactive compound, is commonly found in various fruits such as *Lemons*, *Grapefruits*, *Oranges*, and *Tangerines*. Hst consists of a glycoside aglycone (hesperetin) and a disaccharide (rutinose) [29]. Its antioxidant, anti-inflammatory, cardioprotective, anti-atherosclerotic, and anti-hyperlipidemic properties have been extensively studied. Recently, Hst has also demonstrated anti-coronavirus activity [30,31,32,33]. In this study, we show that Hst effectively inhibits PRRSV infection in Marc-145 and PAM cells in a dose-dependent manner in vitro. Furthermore, our findings indicate that Hst’s anti-PRRSV activity is associated with the downregulation of the P38/JUN/FOS pathway downstream of the P38 MAPK pathway. To our knowledge, this is the first report of Hst’s anti-PRRSV activity and its underlying mechanism.

## 2. Materials and Methods

### 2.1. Cells and Viruses

Marc-145 cells and PAM cells were both obtained from Guangdong Wen’s Food Group Co., Ltd. (Yunfu, Guangdong, China). Marc-145 cells were cultured in Dulbecco’s Modified Eagle’s Medium (DMEM, Gibco, NY, USA) supplemented with 10% fetal bovine serum (FBS, PAN-seratech, Passau, Germany) and subcultured in a 5% CO_2_, 37 °C incubator. PAM cells were cultured in RPMI-1640 medium (Gibco, NY, USA) containing 20% FBS and maintained in a 5% CO_2_, 37 °C incubator for primary culture. The maintenance medium consisted of either DMEM with 2% FBS or RPMI-1640 with 10% FBS. Both complete and maintenance media were supplemented with 100 U/mL penicillin and 100 U/mL streptomycin.

The highly pathogenic PRRSV strain JXA1 [7] was obtained from Guangdong Wen’s Food Group Co., Ltd. (Yunfu, Guangdong, China) and propagated in Marc-145 cells cultured in DMEM maintenance medium containing 2% FBS. Briefly, when Marc-145 cells reached 100% confluence, the JXA1 strain was inoculated and cultured at 5% CO_2_, 37 °C for 2–3 days. Cytopathic effects (CPE) were monitored daily under a microscope, and when CPE exceeded 80%, the culture supernatant was collected. The JXA1-infected cells were subjected to one freeze–thaw cycle, followed by centrifugation at 8000 rpm for 3 min. The supernatant was aliquoted and stored at −80 °C. Finally, the virus titer of JXA1 was determined using the Reed-Muench method [34] and expressed as the tissue culture infective dose 50% (TCID_50_).

### 2.2. Preparation of Hst and Chemicals

Hesperetin (Hst), with a purity of ≥95%, was obtained from Shanghai Macklin Biochemical Technology Co., Ltd. (Shanghai, China) (CAS number: 69097-99-0). Ribavirin (Rib) [35], a broad-spectrum antiviral agent, was used as the positive control for anti-PRRSV activity. Rib was purchased from Shanghai Macklin Biochemical Technology Co., Ltd. (CAS number: 36791-04-5), with a purity of ≥98%. Hst was dissolved in DMSO (Thermo Fisher Scientific, Waltham, MA, USA) and diluted to the working concentration in the maintenance medium, with the final DMSO concentration in the medium remaining below 0.05%. Rib was also diluted to the working concentration in the maintenance medium and filtered through a 0.22 μm sterilizing filter prior to use.

### 2.3. Cytotoxicity Assay

The cytotoxicity of Hst on Marc-145 and PAM cells was assessed using the Cell Counting Kit-8 (CCK-8, Bio-sharp, Hefei, Anhui, China). Briefly, 100 μL of PAM or Marc-145 cell suspensions (approximately 300,000 cells per well for PAM and 50,000 cells per well for Marc-145) were seeded in a 96-well plate and incubated in a 37 °C, 5% CO_2_ incubator. After 12 or 24 h, 100 μL of Hst (200, 100, 50, 25, 12.5, 6.25 μg/mL) dilutions at various concentrations, prepared in the corresponding maintenance medium, were added to each well. A blank control group was also included and incubated further. After 24 or 48 h, 10 μL of CCK-8 solution was added to each well, and the plate was incubated for an additional 2 h. Absorbance at 450 nm was measured using a microplate reader. Cell viability was calculated using the following formula: Cell viability = [(OD (drug) − OD (blank))/(OD (0 drug) − OD (blank))] × 100%.

### 2.4. Antiviral Activity Assay

Marc-145 or PAM cells were seeded into 24-well plates (1 mL/well). After reaching 100% confluence (Marc-45 cells typically took approximately 20–24 h to reach full confluence), the cells were infected with the PRRSV JXA1 strain at a concentration of 100 TCID_50_/mL (1 mL/well). A blank control group and a PRRSV-positive control group were also included and incubated at 37 °C in a 5% CO_2_ incubator. After 2 h, the virus-containing supernatant was discarded, and the cells were washed three times with a serum-free medium. A maintenance medium containing different concentrations of Hst (200, 100, 50, 25, 12.5, 6.25 μg/mL) in DMEM or RPMI-1640 was then added. After 48 h, an indirect immunofluorescence assay (IFA) was performed. At designated time points, post-infection, cells and supernatants were collected for protein analysis (Western blot), virus titer determination, total RNA extraction, and qRT-PCR analysis.

### 2.5. Indirect Immunofluorescence Assay (IFA)

After 48 h of treatment with Hst and PRRSV, Marc-145 or PAM cells were removed from the incubator, washed with PBS, and treated with 0.25% Triton-X 100 solution (500 μL/well) at room temperature for 10 min. The cells were then washed with PBS and incubated with 2% bovine serum albumin (BSA) at room temperature for 1 h. Subsequently, PRRSV monoclonal antibody (1:400 dilution; Beijing Jinnuo Baitai Biotechnology, Beijing, China) was added (500 μL/well) and incubated overnight at 4 °C. After 8–12 h, the cells were washed three times with PBS, each wash lasting 5 min. Goat Anti-Mouse IgG H&L (Alexa Fluor^®^ 568) (1:1000 dilution; Abcam, Cambridge, UK) was then added (500 μL/well) and incubated in a dark, humidified chamber at 37 °C for 1 h. The cells were washed with PBS for 5 min after the incubation. Finally, the cells were incubated in the dark with 4,6-diamidino-2-phenylindole (DAPI, 300 nM) solution at room temperature for 5 min. The cells were then observed under a microscope (KEYENCE BZ-X800, Osaka, Japan) and photographed for documentation.

### 2.6. RNA Extraction and Real-Time PCR Analysis

Total RNA was extracted from the cells using TRIzol reagent (Thermo Fisher Scientific, Waltham, MA, USA) according to the manufacturer’s instructions. RNA was reverse transcribed into first-strand cDNA using a reverse transcription kit (TaKaRa, Dalian, China). Quantitative PCR was performed using FastStart™ Universal SYBR^®^ Green Master Mix (ROX) (Merck, Darmstadt, Germany) on an Applied Biosystems real-time PCR system (Thermo Fisher Scientific, Waltham, MA, USA). All samples were tested in triplicate on the same plate, and the amplification products were analyzed using the comparative threshold cycle (Ct) method. The mRNA expression level of the N gene was normalized to that of the β-actin gene. The primer sequences are provided in Table 1.

### 2.7. Western Blot Analysis

PRRSV-infected cells treated with Hst and control cells were lysed in RIPA buffer (containing 1% protease inhibitors, Beyotime, Shanghai, China), followed by centrifugation to collect the supernatant. The protein concentration was then adjusted to equal levels. SDS loading buffer was added, mixed, and the samples were boiled at 95 °C for 10 min. Equal amounts of protein were separated by 15% SDS-PAGE (Beyotime, Shanghai, China) and transferred to a PVDF membrane (Millipore, NJ, USA). The membrane was blocked in 5% non-fat milk/TBST solution and washed with TBST. It was then incubated overnight at 4 °C with primary antibodies against PRRSV (1:1000, Beijing Jinnuo Baitai Biotechnology, Beijing, China), JUN, FOS, P38, p-P38, and β-actin (1:1000, Wuhan Servicebio Technology Co., Wuhan, Hubei, China). Following washing with TBST, the membrane was incubated with HRP-conjugated goat anti-mouse or anti-rabbit IgG (1:5000, Wuhan Servicebio Technology Co., Wuhan, China) at room temperature in the dark. Protein bands were detected using an enhanced chemiluminescence (ECL) kit (Beyotime, Shanghai, China), and images were captured using a precooled LICOR Odyssey system (LICOR, CT, USA).

### 2.8. Time Course Analysis of Hst Anti-PRRSV

To determine which stage of the PRRSV life cycle is affected by Hst, a time-of-addition experiment was conducted. Marc-145 cells were seeded in 24-well plates and treated with Hst either before, during, or after PRRSV infection. Briefly, when the cells reached 100% confluence, the experiment was initiated at −2 h. In the pre-treatment group, at 0 h, the culture medium was replaced with DMEM maintenance medium, and cells were infected with the PRRSV JXA1 strain (100 TCID_50_/mL, 1 mL/well). Both a cell control group and a positive control group were included. In the co-treatment group, at 0 h, different concentrations of Hst were mixed with PRRSV (100 TCID_50_/mL) and added to the cells (1 mL/well). After 4 h, the culture medium was replaced with DMEM maintenance medium. In the post-treatment group, at 0 h, the cells were infected with PRRSV JXA1 strain (100 TCID_50_/mL, 1 mL/well), and after 4 h, the culture medium was replaced with DMEM maintenance medium containing various concentrations of Hst. All groups were incubated for 48 h.

### 2.9. Direct Inactivation Activity of Hst on Virus Particles

The 200 TCID_50_/mL PRRSV JXA1 strain was mixed with equal volumes of different concentrations of Hst and incubated at 37 °C with 5% CO_2_ for 3 h. After treatment, virus titers were determined using the Reed-Muench method, and the viral genome was detected by qRT-PCR with specific primers, as described above.

### 2.10. Comparative Transcriptomic Analysis

PRRSV-treated Marc-145 cells were collected at 24 (At this time, we observed significant effects on the host cell’s transcriptome, reflecting the immediate impact of Hst on cellular responses to viral infection) hours post-treatment with or without Hst for transcriptomic analysis, with three biological replicates per group. RNA extraction and sequencing were conducted by Baimike Biotechnology Co., Ltd. (Beijing, China). cDNA libraries were sequenced on the Illumina HiSeq™ X-ten platform at Biomarker Technologies Co., Ltd. (Beijing, China), generating 2 × 150 bp paired-end reads according to the standard Illumina protocol. Briefly, total RNA was extracted using TRIzol reagent (Invitrogen, NJ, USA), and its purity and concentration were assessed using a NanoDrop 2000 spectrophotometer. RNA integrity was evaluated using the Agilent 2100/LabChip GX (Agilent, Santa Clara, CA, USA). Once the samples met quality standards, a library was constructed by enriching eukaryotic mRNA using Oligo(dT)-coated magnetic beads. mRNA was randomly fragmented using Fragmentation Buffer, and both the first and second cDNA strands were synthesized and purified. The purified double-stranded cDNA underwent end repair, A-tailing, and adapter ligation, followed by size selection with AMPure XP beads. The cDNA library was amplified by PCR. After library construction, quantification was performed using the Qubit 3.0 fluorometer, with a required concentration of >1 ng/μL. The Qsep400 high-throughput analysis system was used to detect the insert fragments of the library. Once the insert fragments met the expected size, Q-PCR was used to accurately quantify the effective concentration of the library (effective concentration > 2 nM) to ensure quality. After passing quality control, sequencing was conducted using the Illumina NovaSeq6000 platform (Illumina, San Diego, CA, USA) in PE150 mode. Following sequencing, bioinformatics analysis was performed using the BMK Cloud analysis pipeline (www.biocloud.net), accessed on 12 September 2024. On this platform, differential expressi The DESeq2 R package is a method in the BMK Cloud analytics pipeline, and I’ve annotated this information in the description. on analysis was conducted using the DESeq2 R package on two conditions/groups, each with three biological replicates. The *p*-values were adjusted using the Benjamani–Hochberg method to control the false discovery rate. Differentially expressed genes were identified as those with adjusted *p*-values < 0.05. Finally, the effects of Xn on cellular biological processes, molecular functions, and cellular components were examined using GO and KEGG analyses.

### 2.11. siRNA Assay

Marc-145 cells were seeded in a 24-well plate and transfected with 5 µL of Lipofectamine RNAi MAX transfection reagent (Sangon Biotech, Shanghai, China), along with 50 pmol of either siNC (negative control) or siJUN. After 36 h, the cells were infected with 1 mL of PRRSV (100 TCID_50_/mL). Cells were harvested 24 h post-infection for Western blot analysis and qRT-PCR. The sequences of the siRNAs targeting JUN were: (a) 5′-AGACGAGCAGGAGGGCTT-3′ and (b) 5′-AGAACACGCTGCCCAGCG-3′.

### 2.12. The Effect of Hst on the P38/JUN/FOS Signaling Pathway in PRRSV-Infected Cells

Marc-145 cells were seeded in 24-well plates and infected with PRRSV at 100 TCID_50_ when the cell confluence reached 100%. After 2 h of infection, the medium was replaced with DMEM containing various concentrations of Hst, and both a cell control and a positive control group were established. After 24 h, cells were lysed, and the protein and mRNA levels of PRRSV, FOS, JUN, and β-actin were assessed using the methods described above, along with the protein levels of P38 and p-P38.

### 2.13. Statistical Analysis

All statistical analyses were conducted using GraphPad Prism (version 7.0, GraphPad Software, San Diego, CA, USA), and the data are presented as the mean ± standard error of the mean (SEM). The significance of differences among groups was determined using one-way or two-way analysis of variance (ANOVA). Differences with *p*-values < 0.05 were considered significant and are indicated with an asterisk (*) in the figures.

## 3. Results

### 3.1. In Vitro Inhibition of PRRSV Infection by Hst

To evaluate the cytotoxicity of Hst, we performed a CCK-8 assay to assess the effects of various Hst concentrations (Figure 1) on Marc-145 and PAM cells at 24 and 48 h. The results, shown in Figure 1B,C, indicated that cells incubated with Hst concentrations ranging from 6.25 μg/mL to 25 μg/mL for both 24 and 48 h exhibited near 100% relative survival compared to the control group. After 48 h, treatment with 50 μg/mL Hst reduced the survival rate to 90% in Marc-145 cells (*p* < 0.001) and 80% in PAM cells (*p* < 0.001). Based on these findings, 25 μg/mL was selected as the maximum non-cytotoxic concentration of Hst for use in subsequent experiments.

To determine the effective dosage range of Hst for anti-PRRSV activity, Marc-145 and PAM cells were infected with PRRSV (100 TCID_50_/mL) for 2 h, followed by treatment with Hst concentrations of 6.25 μg/mL, 12.5 μg/mL, and 25 μg/mL. After 48 h, antiviral activity was assessed using a PRRSV N protein-specific monoclonal antibody via indirect immunofluorescence assay (IFA). As shown in Figure 2A,F, the number of infected cells in the Hst-treated groups was significantly lower than that in the negative control group, demonstrating dose-dependent inhibition of PRRSV replication. Viral titers and PRRSV N mRNA levels were assessed using TCID_50_ and qRT-PCR assays at 24 and 48 h post-infection. The results indicated a dose-dependent reduction in viral titers and N mRNA levels at each time point (Figure 2D,E,I,J). To further confirm the anti-PRRSV effects of Hst, Western blot analysis was performed using antibodies specific to viral N protein and β-actin to examine N protein levels in PRRSV-infected cells at 24 and 48 h. As shown in Figure 2B,C,G,H, both N protein and mRNA levels decreased gradually with increasing Hst concentrations. These results collectively indicate that Hst exerts anti-PRRSV activity in vitro. Ribavirin (Rib), a well-known viral RNA polymerase inhibitor, was used as a positive control antiviral agent in this study. As expected, treatment with 50 μg/mL Rib significantly inhibited PRRSV replication in both Marc-145 and PAM cells.

### 3.2. Inhibition of PRRSV Infection by Different Treatment Protocols of Hst

To investigate the mechanism underlying Hst-mediated inhibition of PRRSV, Marc-145 cells were treated with Hst prior to (pre-treatment), during (co-treatment), or after (post-treatment) PRRSV infection (Figure 3A). After 48 h, immunofluorescence assay (IFA) analysis was performed, and cell lysates were collected for RT-PCR detection. As shown in Figure 3B,C, pre-treatment with Hst for 2 h did not significantly reduce the expression of the viral N protein, and RT-PCR results were similar to those of the PRRSV control group. This suggests that pre-treatment with Hst did not notably inhibit PRRSV replication in Marc-145 cells. In contrast, co-treatment with Hst during PRRSV infection resulted in a significant reduction in viral N protein expression at a concentration of 25 μg/mL (*p* < 0.05), while lower concentrations (6.25 μg/mL and 12.5 μg/mL) showed no effect (Figure 3B,D). These findings indicate that co-treatment with Hst during infection moderately inhibits PRRSV replication. Interestingly, post-treatment with Hst following PRRSV infection led to a dose-dependent inhibition of viral N protein expression, with concentrations ranging from 6.25 μg/mL to 25 μg/mL, significantly reducing PRRSV replication (Figure 3B,E). In conclusion, these results demonstrate that Hst treatment during or after viral infection significantly inhibits PRRSV replication in Marc-145 cells, with post-infection treatment being more effective than co-treatment.

### 3.3. Hst Does Not Directly Interact with PRRSV

To assess whether Hst can directly neutralize PRRSV particles, we incubated PRRSV with various concentrations of Hst in DMEM medium at 37 °C for 3 h. Following incubation, the virus was diluted 1000-fold to minimize any potential effects of the drug on the virus, and Marc-145 cells were reinfected. After 48 h, we performed PRRSV mRNA analysis, immunofluorescence assay (IFA), and TCID50 assays to evaluate the infectivity of treated PRRSV. As shown in Figure 4A–C, incubation with Hst (6.25, 12.5, or 25 μg/mL) did not reduce the ability of PRRSV to infect Marc-145 cells, indicating that Hst does not directly interact with PRRSV particles. Similarly, the positive control drug, Rib (50 μg/mL), did not affect PRRSV infectivity.

### 3.4. Transcriptional Response of PRRSV-Infected Cells to Hst Treatment

Hst has been shown to inhibit neuroinflammation in microglia by reducing inflammatory cytokines and blocking the MAPK pathway. It also regulates osteoclast differentiation through MAPK pathway inhibition and reduces cartilage damage by targeting the P38 MAPK pathway, as previously reported [36,37,38]. To investigate the mechanism underlying Hst’s anti-PRRSV effects, we performed high-throughput RNA sequencing to analyze the gene expression profiles of Marc-145 cells treated with either Hst or DMEM for 24 h, as well as PRRSV-infected Marc-145 cells treated with either Hst or DMEM for 24 h. The results are presented in Figure 5. In Figure 5A, the number of differentially expressed genes (DEGs) for each group is shown, using a *p*-value threshold of <0.05. Genes are considered upregulated if their expression is higher in Hst-treated Marc-145 cells compared to those treated with DMEM, and downregulated if the expression shows the opposite trend. The data indicate that the number of DEGs was significantly higher in PRRSV-infected Marc-145 cells treated with either Hst or DMEM compared to the control groups. In the control group of Marc-145 cells treated with either DMEM or Hst, 1364 genes were upregulated and 1886 genes were downregulated. In the control group of PRRSV-infected Marc-145 cells treated with DMEM, 94 genes were upregulated and 65 genes were downregulated. In the group of PRRSV-infected Marc-145 cells treated with Hst or DMEM, 1318 genes were upregulated and 1899 genes were downregulated (Figure 5A). A total of 25 common DEGs were identified across the three comparison groups (Figure 5B–E).

### 3.5. GO and KEGG Pathway Analysis

Kyoto Encyclopedia of Genes and Genomes (KEGG) enrichment analysis is used to assess whether differentially expressed genes (DEGs) are significantly associated with specific biological pathways. By applying enrichment methods, the most relevant signaling or metabolic pathways involving the DEGs can be identified [39]. To further elucidate the anti-PRRSV mechanism of Hst, we performed KEGG pathway enrichment analysis on Marc-145 cells treated with either Hst or DMEM for 24 h, and on Marc-145 cells infected with PRRSV and then treated with Hst or DMEM 24 h post-infection. The top 20 most significant pathways were selected for analysis. The results revealed that the MAPK signaling pathway was enriched in all three groups. Notably, in the PRRSV-infected group treated with DMEM for 24 h, genes associated with the MAPK signaling pathway were significantly upregulated (red boxed area in Figure 6B), indicating that PRRSV infection activates the MAPK signaling pathway. In contrast, in the other comparison groups, genes related to the MAPK pathway were both upregulated and downregulated (red boxed areas in Figure 6A,C), suggesting that Hst treatment modulates the MAPK signaling pathway in Marc-145 cells.

Gene Ontology (GO) enrichment analysis aims to identify differentially expressed genes (DEGs) based on experimental objectives and examine their distribution across various GO categories to analyze functional differences between experimental groups [40]. In this study, Goseq software (www.biocloud.net, accessed on 12 September 2024) was used to perform GO enrichment analysis on the DEGs from three comparison groups. The distribution and counts of enriched DEGs in biological processes, cellular components, and molecular functions were visually presented, with the top 20 most significant GO terms selected. The results revealed that, in all three comparison groups, biological processes involved genes related to the MAPK signaling pathway or its sub-pathways (red box areas in Figure 7A–C). In the DMEM vs. Hst-treated Marc-145 cells and DMEM vs. PRRSV-treated Marc-145 cells comparison groups, molecular functions were associated with genes in the MAPK signaling pathway or its sub-pathways (red box areas in Figure 7A,B). Furthermore, in the PRRSV-treated Marc-145 cells vs. DMEM-treated Marc-145 cells comparison group, genes related to the MAPK signaling pathway in both biological processes and molecular functions were upregulated (red box area in Figure 7B). In the other two comparison groups, genes related to the MAPK signaling pathway showed both upregulation and downregulation in biological processes and molecular functions (red box areas in Figure 7A,C).

Given that PRRSV can upregulate MAPK pathway-related genes in Marc-145 cells and based on the results of KEGG and GO analyses, we hypothesize that the anti-PRRSV mechanism of Hst may involve the regulation of MAPK signaling pathway-related genes. Further investigation revealed that Hst treatment in PRRSV-infected cells led to the downregulation of several MAPK pathway-related genes, including NR4A1, JUN, FOS, DUSP1, and DUSP10 (Figure 8A). These genes are integral to the P38 MAPK signaling pathway and play key roles in inflammation, apoptosis, and cellular growth [41].

To validate the results of the transcriptomic analysis, RT-PCR was performed to assess the expression of NR4A1, JUN, FOS, and DUSP1 genes in Marc-145 cells treated with either DMEM or PRRSV for 24 h, as well as in cells treated with Hst or DMEM after PRRSV infection for 24 h. The results shown in Figure 8B–E demonstrate that, compared to the DMEM group, Hst treatment significantly reduced the expression of NR4A1, JUN, FOS, and DUSP1 induced by PRRSV. Interestingly, Hst treatment also increased the expression of JUN, FOS, and DUSP1 compared to the DMEM group. These findings are consistent with the transcriptomic data and suggest that Hst may exert a dual regulatory effect on genes involved in the P38 MAPK pathway. To further explore the P38 MAPK-related network, a protein interaction network was generated for the P38 MAPK pathway genes. As shown in Figure 8F–H, the network identified interactions between JUN, FOS, and DUSP1 across all treatment groups, indicating that Hst regulates proteins within this network. Overall, these data suggest that Hst may protect cells from PRRSV infection by downregulating the P38 MAPK signaling pathway, potentially involving the P38/JUN/FOS axis.

### 3.6. Hst Inhibits PRRSV Proliferation by Downregulating the P38/JUN/FOS Pathway

The transcriptomic analysis revealed alterations in the expression of both FOS and JUN proteins across the three comparison groups. PRRSV infection upregulated the expression of FOS and JUN, while Hst treatment significantly reduced their expression [42]. To further investigate the role of Hst in suppressing FOS and JUN expression during PRRSV infection, Marc-145 cells infected with PRRSV were treated with various concentrations of Hst. Western blotting was performed to assess the levels of PRRSV, FOS, and JUN proteins. As shown in Figure 9A, Hst demonstrated a dose-dependent inhibitory effect on PRRSV, FOS, and JUN expression, indicating that Hst significantly suppresses both FOS and JUN proteins. The results from PRRSV TCID_50_ assays, RT-PCR, and qPCR for FOS and JUN expression were consistent with the protein data (Figure 9B–E).

P38, an upstream regulator in the JUN/FOS cascade, is a signature protein of the P38 MAPK pathway. To determine whether Hst downregulates the P38 MAPK pathway in PRRSV-infected cells, Marc-145 cells were treated with different concentrations of Hst, and Western blotting was used to evaluate the expression of PRRSV, P38, and p-P38 proteins. As shown in Figure 9F, Hst treatment inhibited the expression of PRRSV and p-P38 proteins in a dose-dependent manner, whereas the expression of total P38 remained unaffected. These findings suggest that PRRSV infection promotes the expression of FOS, JUN, and p-P38 proteins, thereby enhancing viral replication. After Hst treatment, the expression of these proteins was significantly reduced, further confirming the involvement of FOS, JUN, and p-P38 in Hst’s anti-PRRSV effects in Marc-145 cells.

JUN is an upstream regulator of FOS. To investigate the role of JUN in PRRSV infection, we designed two siRNAs targeting the JUN gene using monkey and pig-derived JUN sequences as templates. Marc-145 cells were cultured in 24-well plates, and each well was transfected with 50 pmol of siJUN-a, siJUN-b, or siNC. After 24 h of transfection, the cells were infected with PRRSV at a titer of 100 TCID_50_. At 36 h post-infection, cells were collected for Western blotting and qRT-PCR analysis. The results indicated that transfection with the siJUN-b fragment led to a more significant reduction in both mRNA (Figure 10B) and protein levels (Figure 10A) compared to the other treatments. Therefore, siJUN-b was selected for further experiments.

To investigate the role of JUN in Hst-mediated anti-PRRSV activity, Marc-145 cells were transfected with siJUN-b or siNC. After 24 h, the cells were infected with PRRSV at a titer of 100 TCID_50_, and 2 h post-infection, they were treated with 25 µg/mL Hst (Xn) or DMEM. Cells were collected at 36 h for qRT-PCR and Western blotting. The results showed that, in siNC-transfected cells, Hst significantly reduced both the mRNA levels of the PRRSV N protein (*p* < 0.001) and the expression of N protein. In siJUN-transfected cells, Hst still significantly reduced the mRNA levels of PRRSV N protein (*p* < 0.01) (Figure 10C,D). To assess whether JUN has a hierarchical regulatory effect on downstream FOS and upstream P38 proteins, we performed qRT-PCR and Western blotting on cells collected at 36 h. The results revealed that in siNC-transfected cells, as JUN expression decreased, the expression of p-P38 protein also decreased, leading to a significant reduction in FOS and PRRSV N protein levels, as well as their corresponding mRNA levels (*p* < 0.001). These findings suggest that there is a cascade regulatory effect between P38/JUN/FOS, with JUN acting as an effector that promotes PRRSV proliferation. Furthermore, Hst’s anti-PRRSV effect may be mediated by its regulation of the P38/JUN/FOS pathway.

## 4. Discussion

Natural products have long been recognized as a rich source of medicinal compounds, with the therapeutic use of plants dating back thousands of years [24,27]. These natural compounds also serve as valuable resources for developing antiviral drugs. For example, matrine has been shown to inhibit PRRSV infection both in vitro and in vivo, suppressing PRRSV replication by modulating the expression of HSPA8 and HSP90AB1 [43]. Allicin inhibits HP-PRRSV and NADC30-like PRRSV in a dose-dependent manner by interfering with viral entry, replication, and assembly. Additionally, it restores the pro-inflammatory, TNF signaling, and MAPK signaling pathways that are dysregulated by PRRSV [23]. In this study, we found that Hst significantly inhibits PRRSV proliferation in Marc-145 and PAMs cells by downregulating the P38/JUN/FOS signaling pathway within the P38 MAPK pathway, highlighting its promising therapeutic potential for treating PRRSV infections.

Hesperidin (Hst) is a flavonoid glycoside, a subclass of flavonoids, and a phytoestrogen commonly found in various fruits and food sources such as *Lemons*, *Grapefruits*, *Oranges*, and *Tangerines*. It has been extensively studied for its antioxidant, anti-inflammatory, cardioprotective, anti-atherosclerotic, and anti-hyperlipidemic activities. Although previous studies have primarily focused on its activity against yellow fever virus [44] and influenza virus [45], Hst has recently gained significant attention for its anti-COVID-19 properties. It has been shown to suppress viral infections through multiple mechanisms, including inhibition of viral entry, replication, and modulation of inflammatory responses [29,32,33]. These findings highlight the broad range of biological activities of Hst.

In this study, a 48-h time point was used to investigate the longer-term antiviral effects of Hst. We demonstrated that Hst effectively inhibits PRRSV infection in Marc-145 and PAMs cells in a dose-dependent manner through in vitro cell experiments. One major challenge with antiviral drugs is the development of resistance, where some viral particles survive drug treatment, mutate, and accumulate resistance over time [46]. RNA viruses are more prone to acquiring resistance than DNA viruses due to their higher mutation rates [47]. A significant issue with antiviral drugs is that long-term use can lead to drug resistance. For instance, studies have shown that even low concentrations of ribavirin can induce resistance to PRRSV with prolonged use, as the virus survives and develops resistance through ongoing mutations [35,48]. Although Hst is a natural compound with potent anti-PRRSV effects in vitro, it is important to note that PRRSV is not completely inhibited at a concentration of 25 µg/mL. Therefore, further research is needed to investigate the long-term effects of Hst use on the development of PRRSV resistance.

To explore the underlying mechanisms of Hst’s anti-PRRSV activity, we conducted transcriptome analysis on Hst-treated Marc-145 cells and PRRSV-infected Marc-145 cells treated with Hst. The 24-h time point was chosen to evaluate the early transcriptomic changes induced by Hst treatment. At this time, we observed significant effects on the host cell’s transcriptome, reflecting the immediate impact of Hst on cellular responses to viral infection. This time frame allowed us to capture the early antiviral effects of Hst, particularly in terms of gene expression modulation and early-stage viral replication [24]. The results revealed significant enrichment of MAPK pathway-related genes in these groups, including Hst-treated Marc-145 cells compared to DMEM-treated controls, PRRSV-treated Marc-145 cells compared to DMEM-treated controls, and Hst-treated PRRSV-infected Marc-145 cells compared to PRRSV-infected controls.

The MAPK signaling cascade plays a critical role in regulating cellular processes such as proliferation, differentiation, apoptosis, and stress response [18]. P38 MAPK is a key component of this signaling pathway, responding to various stimuli, including viral infections [14]. Many viral infections activate the P38 MAPK pathway, and inhibiting this pathway has been shown to have significant antiviral effects. For example, HCV infection enhances viral replication by increasing P38 phosphorylation via TAB1 (TGF-β-activated kinase 1) binding. Similarly, infections with SFTSV, HSV-1, and SARS-CoV-2 activate P38, and the use of P38 inhibitors reduces replication of these viruses [15]. In the case of PRRSV infection, activation of the P38 MAPK pathway has been implicated in inflammatory responses, and P38 inhibitors have been shown to significantly reduce viral replication, RNA synthesis, and viral protein expression [21]. Furthermore, PRRSV infection activates the MAPK pathway to upregulate inflammatory cytokine expression, which contributes to lung injury following infection [19,49,50].

Our study observed that Hst dose-dependently reduced the overexpression of JUN and FOS, key downstream factors in the P38 MAPK pathway, induced by PRRSV. The reduction in JUN and FOS expression led to a decrease in PRRSV replication. JUN and FOS are nuclear oncogenes that, when dimerized, form the activator protein 1 (AP-1) complex [51]. AP-1, composed of JUN and FOS family proteins, binds to common DNA motifs, such as the TPA response element (TRE) or AP-1 sites, to regulate gene transcription in response to various extracellular signals, including growth factors, cytokines, tumor promoters, and DNA-damaging agents [52]. In the context of viral infections, HCV enhances the phosphorylation of JUN and FOS by upregulating the double-stranded RNA-activated protein kinase R (PKR), promoting viral replication [53]. Additionally, JUN and FOS bind to AP-1 sites in gene promoters, triggering transcriptional activation of cytokines involved in macrophage differentiation, as well as viral gene expression and progeny virus production in certain viral infections, such as lentivirus and encephalitis viruses [54].

Although numerous studies have demonstrated that JUN and FOS play crucial roles in viral transcription and replication, the exact mechanisms remain unclear. For example, Influenza A Virus (IAV) infection triggers the activation of NLRP3 inflammasomes and an inflammatory response. Knockdown of JUN decreases pro-IL-1β mRNA levels and inhibits NLRP3 inflammasome activation, thereby suppressing IAV-induced inflammation [55]. However, the relationship between JUN, FOS, and PRRSV replication has not been well established. In this study, we show that JUN and FOS are involved in PRRSV infection in Marc-145 cells, and that Hst directly targets the JUN and FOS pathways to mitigate PRRSV infection. Although Hst has been shown to exhibit a strong anti-PRRSV effect in both PAM and Marc-145 cells, transcriptome experiments in this study were performed and validated only in Marc-145 cells. However, both FOS and JUN genes were present in both cell lines, suggesting that the antiviral mechanism of Hst in PAM cells may be related to this pathway.

Interestingly, our findings reveal that Hst exerts a dual regulatory effect on JUN and FOS. It promotes the secretion of JUN and FOS in PRRSV-infected Marc-145 cells, while simultaneously inhibiting their overexpression. This suggests that Hst may prepare infected cells for PRRSV suppression while protecting the cells from potential toxicity caused by excessive JUN and FOS expression. These results imply that JUN and FOS may facilitate PRRSV replication in Marc-145 cells by enhancing viral transcription.

We also investigated the expression and phosphorylation of P38, an upstream regulator in the JUN and FOS pathways. Hst treatment reduced P38 phosphorylation in PRRSV-infected Marc-145 cells without affecting P38 expression. Additionally, knocking down JUN expression led to a decrease in P38 phosphorylation, suggesting an inverse regulatory relationship between JUN and P38 phosphorylation. Since JUN is a key component of the AP-1 complex, these findings indicate that JUN plays a role in regulating the activation of P38 in PRRSV-infected cells.

While our study provides insights into the complex signaling interactions between JUN, FOS, and P38, further experiments are needed to elucidate the detailed regulatory network involved in PRRSV infection and to identify the specific molecular targets of Hst. Previous research has shown that the P38 MAPK pathway is critical for regulating inflammation, particularly through pathways like NF-κB. Inhibiting the P38 MAPK pathway has been shown to modulate the overexpression of inflammatory cytokines triggered by viral infections or LPS exposure [56,57,58]. Early studies suggest that NF-κB activation is a typical response to PRRSV infection [59], with PRRSV exhibiting a dual role in regulating NF-κB—suppressing it early in infection and activating it later [60]. Optimal PRRSV replication has been linked to the activation of NF-κB, and inhibiting this pathway can impair viral replication [61,62]. In our study, Hst effectively suppressed the overactivation of the P38 MAPK pathway induced by PRRSV. Further investigations are required to determine whether Hst suppresses the inflammatory response induced by PRRSV and to explore the regulatory relationship between the P38 MAPK pathway and NF-κB.

Adverse toxicological effects can be classified into chemical-based, targeted, or off-target effects. Targeted effects refer to the amplification of pharmacological actions on the intended target within the test system, while off-target effects involve adverse reactions due to the modulation of unintended targets, which may either be biologically relevant or unrelated to the target of interest [63]. For instance, HIV-1 latency reversal agents (LRAs) can activate both classical and non-classical NF-κB signaling pathways, leading to the expression of reverse transcriptase elements as an off-target effect, thereby reactivating HIV-1 [64]. This study shows that HST’s anti-PRRSV mechanism primarily targets the P38/JUN/FOS pathway. Therefore, further studies are necessary to determine whether HST induces adverse toxicological effects, either through targeted or off-target actions, particularly concerning its impact on anti-inflammatory, pro-inflammatory, and antioxidant activities.

The PRRSV lifecycle can be divided into four basic stages: attachment, entry, replication, and release [4,65]. Our results show that HST significantly inhibits PRRSV primarily during the post-treatment stage, which includes PRRSV replication and release. Several host cell-dependent factors can influence and control PRRSV attachment and uptake, including (i) the level of specific viral receptors, such as CD163; (ii) host cell-derived innate immune defense molecules that bind to and neutralize infectious viral particles; and (iii) antiviral mediators that restrict viral replication [66]. Therefore, altering any one or more of these factors through Hst treatment may affect susceptibility to PRRSV infection. The CD163 molecule is a necessary receptor for PRRSV and plays a critical role in determining cell susceptibility to the virus. Studies have shown that pigs with a CD163 gene knockout are resistant to NA-type PRRSV infection [67]. Further research has demonstrated that the SRCR5 domain is essential for viral infection, while pigs with a CD163 SRCR5 gene defect are resistant to NA-type PRRSV strains. In contrast, SRCR1–4 and the cytoplasmic tail are non-essential for viral infection [68,69,70]. Since CD163 is a critical receptor for PRRSV, directly targeting CD163 to identify new compounds against PRRSV infection has shown significant anti-PRRSV effects. For example, *Plantago ovata*, *Rhus chinensis*, and *Persicaria orientalis* exhibited anti-PRRSV effects in vitro [71]. These compounds mainly target CD163 to affect PRRSV attachment and entry. Our study found that Hst primarily inhibits PRRSV during the replication and release stages. Therefore, future studies could explore combination therapies with these compounds. Targeting viral replication through pathways such as P38/JUN/FOS may provide an additional approach, particularly when immune responses need to be regulated to reduce viral load and prevent inflammation.

We found that HST is active against PRRSV replication at concentrations ranging from approximately 25 to 6.25 μg/mL in vitro, raising the question of whether such concentrations could inhibit PRRSV in pigs. Many of the active anti-PRRSV compounds identified in vitro will be further validated in pigs. Xanthohumol, an isoprenylated flavonoid extracted from hops (*Humulus lupulus* L.), has been shown to inhibit PRRSV replication in vitro at concentrations of 5–25 µM [24]. In vivo, pigs treated with 25 mg/kg exhibited the fewest clinical symptoms, closely resembling those in the mock-negative group [72]. Therefore, further research is needed to evaluate the effect of Hst against PRRSV in vivo. Additionally, by measuring plasma drug concentrations under different treatment modalities (oral or injection), we can determine whether the plasma concentration in pigs reaches 6.25 µg/mL. If this concentration is achieved, Hst has the potential to exert an anti-PRRSV effect in infected pigs.

## 5. Conclusions

In summary, our study demonstrates that Hst suppresses PRRSV replication in a dose-dependent manner. Transcriptomic analysis revealed that Hst downregulates the MAPK pathway activated by PRRSV infection. The anti-PRRSV activity of Hst appears to be primarily mediated through inhibition of the P38/JUN/FOS pathway, a critical regulator of the P38 MAPK signaling cascade. These findings provide valuable insights into the molecular mechanisms underlying Hst’s antiviral effects and offer promising therapeutic potential for combating PRRSV-induced infections.

## Figures and Tables

**Figure 1 microorganisms-13-00450-f001:**
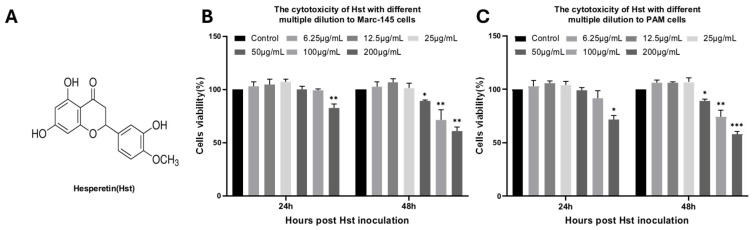
Cytotoxicity of Hst. (**A**) Chemical structure of Hst. (**B**,**C**) Cytotoxicity of Hst assessed by CCK-8 assay in Marc-145 and PAM cells after 24 and 48 h of incubation. *n* = 3. Bars represent means ± SEMs from three independent experiments. *** *p* < 0.001; ** *p* < 0.01; * *p* < 0.05.

**Figure 2 microorganisms-13-00450-f002:**
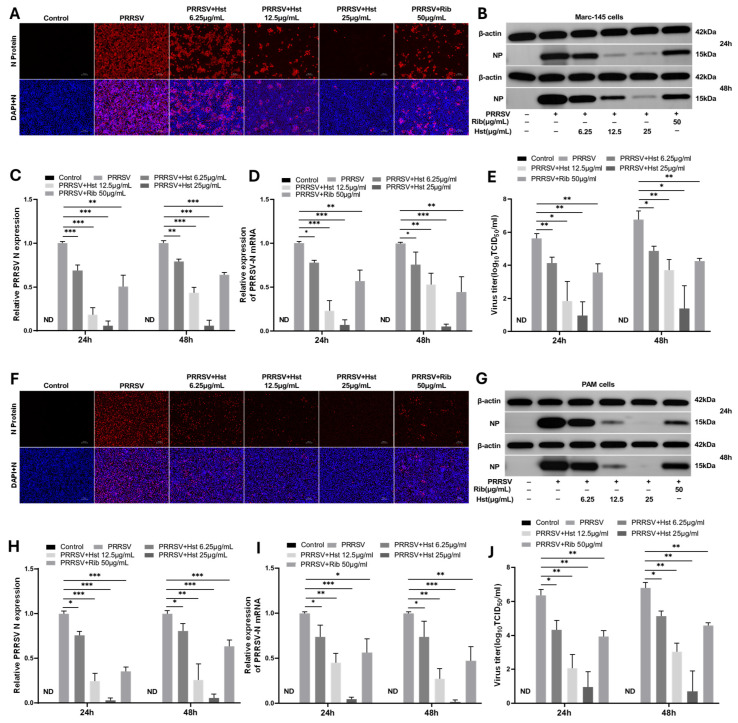
Inhibition of PRRSV replication by Hst in vitro. (**A**,**F**) Immunofluorescence assay (IFA) showing Hst’s antiviral activity against PRRSV infection in Marc-145 and PAM cells at 48 h post-treatment. Cells in a 96-well plate were infected with 100 TCID50/mL PRRSV for 2 h, followed by treatment with DMEM or RPMI-1640 media containing various concentrations of Hst (12.5, 25, or 50 μg/mL). IFA was performed after 48 h. Ribavirin (Rib) was used as a positive control. Red indicates PRRSV N protein, and blue represents the cell nucleus. (**B**,**G**) Western blot analysis of PRRSV N protein expression in cells infected with 100 TCID_50_/mL PRRSV for 2 h, followed by treatment with Hst at different concentrations. Ribavirin served as a positive control. (**C**,**H**) Quantification of Western blot results, with protein levels normalized to β-actin. (**D**,**I**) Relative expression levels of PRRSV N protein, as analyzed by RT-PCR. (**E**,**J**) Viral titers in cell lysates were determined using TCID50 assay. *n* = 3. Bars represent means ± SEMs from three independent experiments. Note: ND (not detected); *** *p* < 0.001; ** *p* < 0.01; * *p* < 0.05. Scale bar, 100 μm.

**Figure 3 microorganisms-13-00450-f003:**
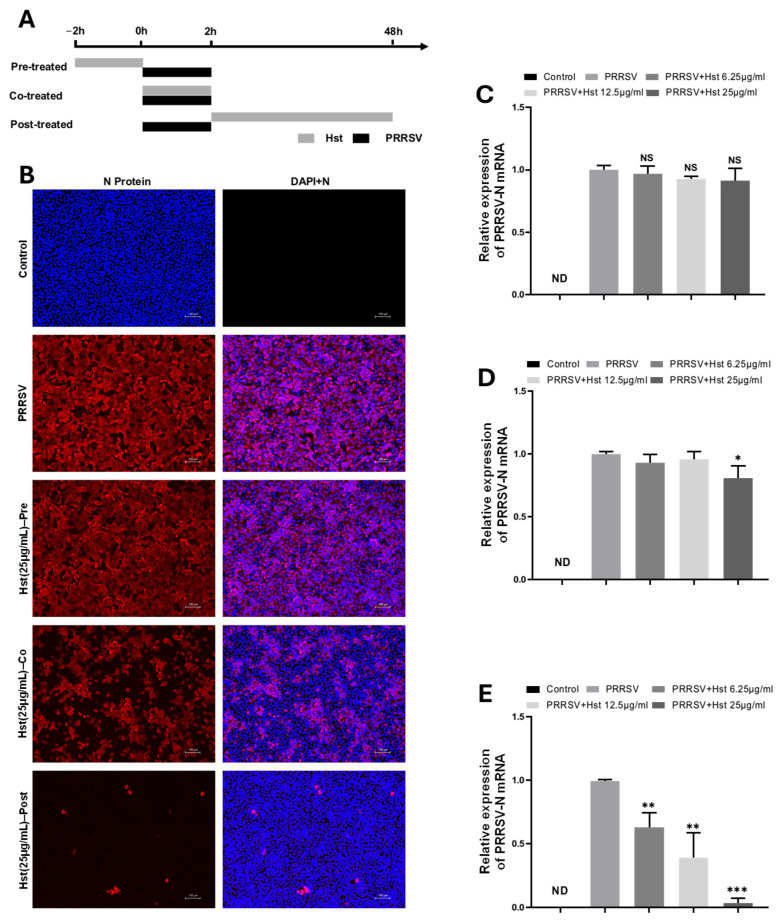
Inhibition of PRRSV replication by Hst in Marc-145 cells under co-treatment and post-treatment conditions. (**A**) Schematic of time-based treatment. Marc-145 cells in 24-well plates were infected with 100 TCID50/mL of PRRSV for 2 h and subsequently treated with different concentrations of Hst (6.25, 12.5, or 25 μg/mL) at different time points: pre-treatment (Pre), co-treatment (Co), and post-treatment (Post). (**B**) Effects of different Hst treatment modes on PRRSV N protein expression were assessed by immunofluorescence assay (IFA), respectively. (**C**–**E**) Effects of different Hst treatment modes on PRRSV N mRNA by RT-PCR, respectively. (**C**) Pre-treatment (Pre), (**D**) Co-treatment (Co), (**E**) Post-treatment (Co). *n* = 3. Bars represent means ± SEMs from three independent experiments. Note: ND (not detected); NS (no significance); *** *p* < 0.001; ** *p* < 0.01; * *p* < 0.05. Scale bar, 100 μm.

**Figure 4 microorganisms-13-00450-f004:**
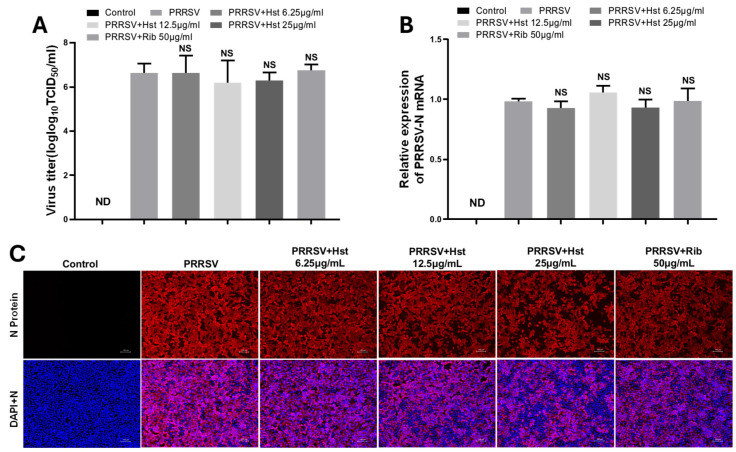
Hst does not directly interact with PRRSV. PRRSV was incubated with different concentrations of Hst (6.25, 12.5, or 25 μg/mL) at 37 °C for 3 h. After incubation, the virus was diluted 1000-fold to minimize any potential effects of the drug on the virus and then reinfected into Marc-145 cells. After 48 h, viral titers were measured by TCID50 (**A**), PRRSV N mRNA expression levels were analyzed by RT-PCR (**B**), and PRRSV N protein was detected by immunofluorescence assay (IFA) (**C**). *n* = 3. Note: ND (not detected); NS (no significance); Bars represent means ± SEMs from three independent experiments. Scale bar, 100 μm.

**Figure 5 microorganisms-13-00450-f005:**
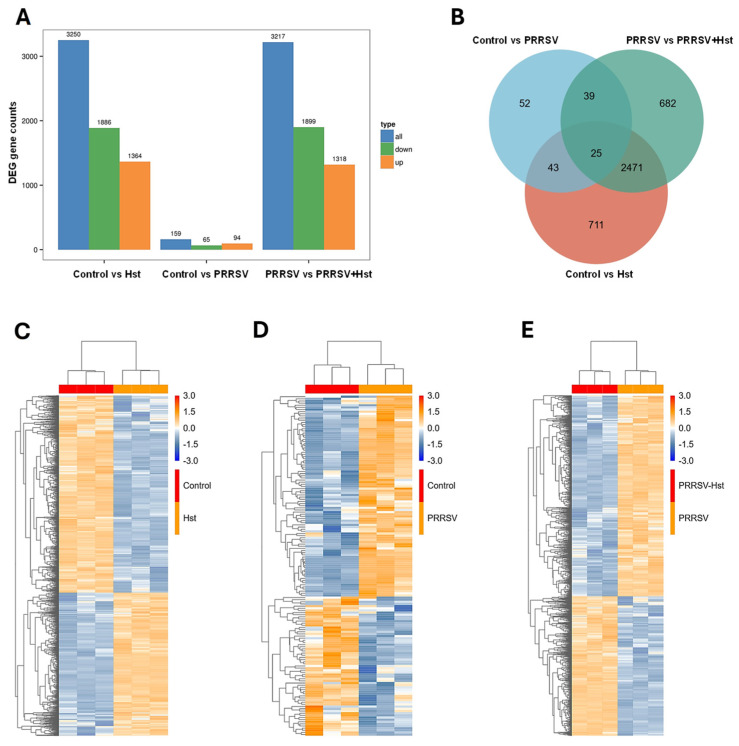
Impact of Hst treatment on the transcriptome of Marc-145 cells. (**A**) RNA sequencing analysis of Marc-145 cells treated with 12.5 μg/mL Hst or DMEM for 24 h, and Marc-145 cells treated with 12.5 μg/mL Hst or DMEM following PRRSV infection for 24 h. Bar graphs show the number of upregulated and downregulated genes for each treatment group, selected based on a fold change ≥ 1.5 and a *p*-value < 0.05 as differential gene screening criteria. (**B**) Venn diagram of differentially expressed genes (DEGs) across the comparison groups. The numbers in each section represent the number of genes in the corresponding category, with overlapping areas indicating the number of common DEGs between groups. (**C**–**E**) Cluster heatmap of DEG expression levels. The *x*-axis represents the sample names and clustering results, while the *y*-axis represents the DEGs and their clustering. Each column corresponds to a different sample (*n* = 3), and each row corresponds to a specific gene. The color scale represents gene expression levels in the samples (log10[FPKM + 0.000001]).

**Figure 6 microorganisms-13-00450-f006:**
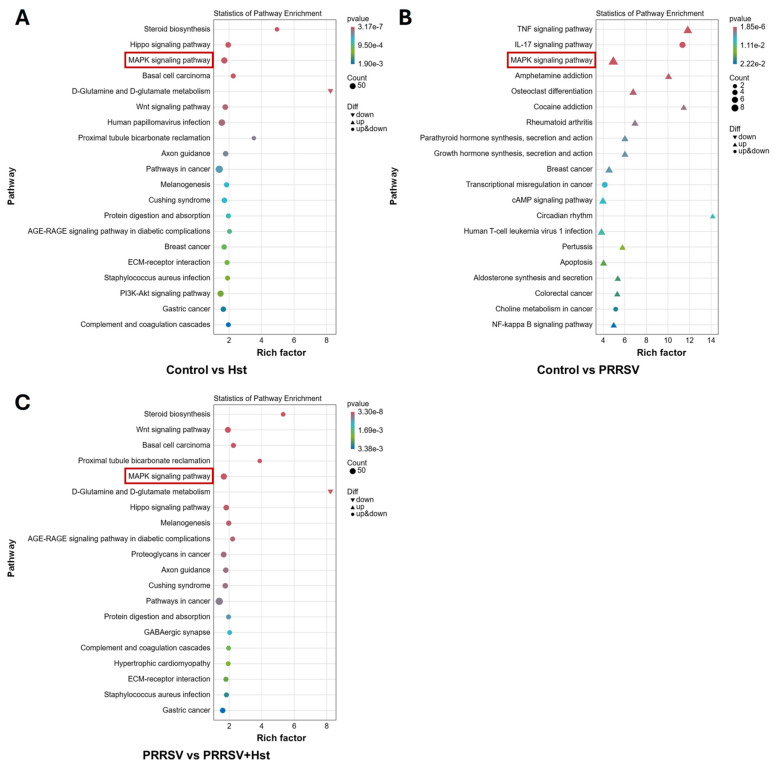
KEGG Pathway Analysis. (**A**–**C**) KEGG pathway analysis for different comparison groups. Each circle represents a KEGG pathway, with the color indicating the *p*-value from the hypergeometric test. The *y*-axis corresponds to the pathway name, while the *x*-axis represents the enrichment factor (Rich factor), defined as the ratio of differentially expressed genes annotated to a specific pathway to the total number of genes annotated to that pathway. Pathways located closer to the upper right corner have higher reference value. Note: The red boxes represent pathways that are common in the three comparison groups.

**Figure 7 microorganisms-13-00450-f007:**
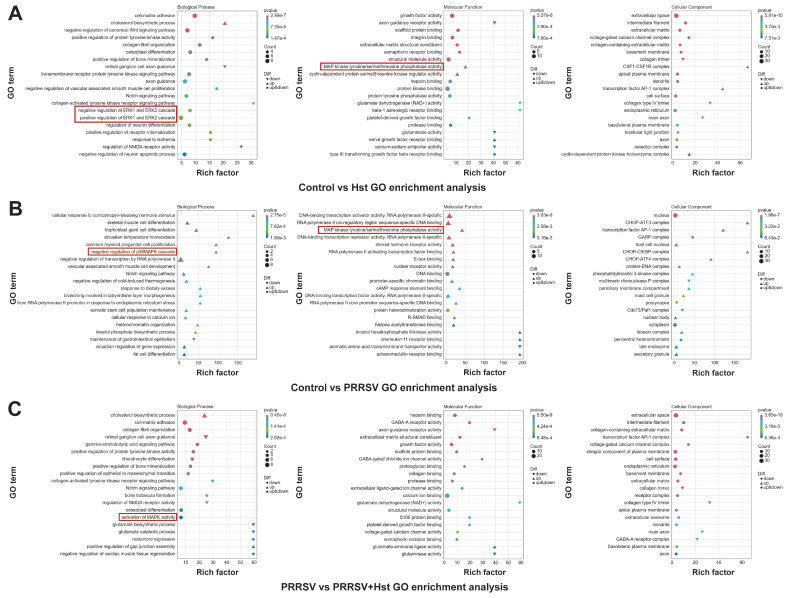
GO Enrichment Analysis. (**A**–**C**) GO enrichment analysis for different comparison groups. The *x*-axis represents GeneNum, which denotes the number of genes of interest annotated in each GO term, while the *y*-axis corresponds to each GO annotation term. The color of each circle indicates the *p*-value from the hypergeometric test. Note: The red box represents MAPK signaling pathway or its sub-pathways.

**Figure 8 microorganisms-13-00450-f008:**
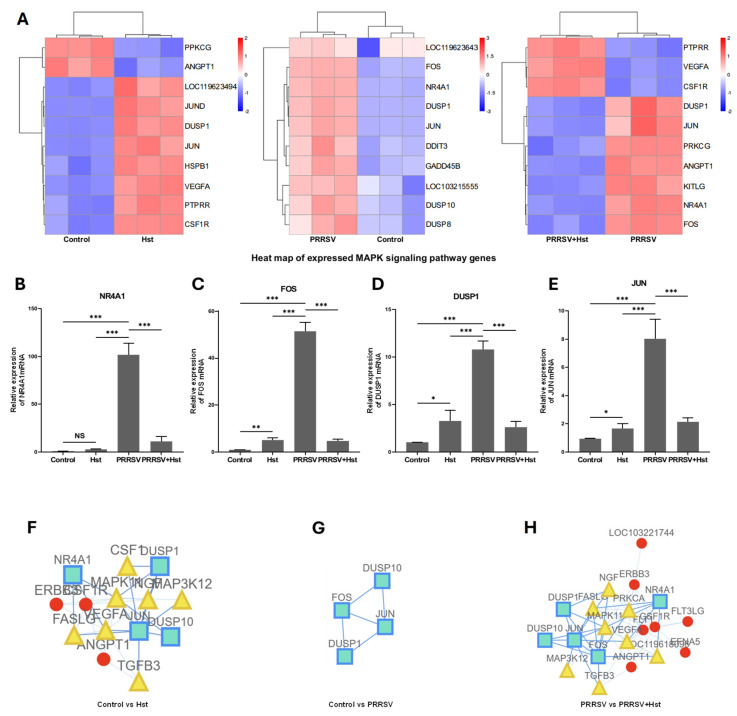
Effect of Hst on MAPK Pathway and Related Genes. (**A**) Impact of DMEM or Hst treatment on MAPK-related genes in Marc-145 cells after 24 h, and in PRRSV-infected Marc-145 cells treated with Hst or DMEM for 24 h. Red and blue indicate upregulated and downregulated genes, respectively. (**B**–**H**) RT-PCR analysis of MAPK-related genes in Marc-145 cells treated with DMEM or Hst for 24 h, and in PRRSV-infected Marc-145 cells treated with Hst or DMEM for 24 h. (**F**–**H**) A protein–protein interaction (PPI) network for differentially expressed MAPK-related genes was constructed using the STRING database (www.biocloud.net, accessed on 12 September 2024). Nodes in the interaction map represent proteins (Gene ID/Gene Name), and edges represent the interaction relationships between proteins. The main network is depicted in blue, while sub-networks are shown in yellow and red. *n* = 3. Bars represent means ± SEMs from three independent experiments. *** *p* < 0.001; ** *p* < 0.01; * *p* < 0.05. Note: NS (no significance).

**Figure 9 microorganisms-13-00450-f009:**
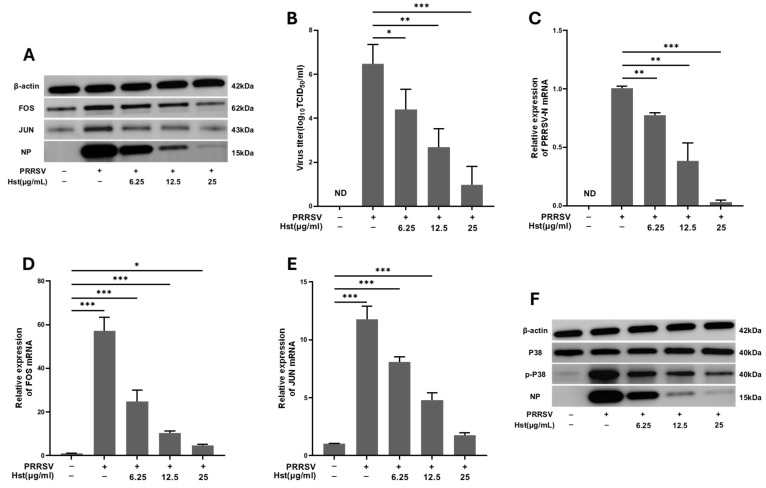
Hst inhibits PRRSV replication in Marc-145 cells by downregulating the P38/JUN/FOS pathway. (**A**) Marc-145 cells were infected with 100 TCID_50_/mL of PRRSV for 2 h and then treated with various concentrations of Hst (6.25, 12.5, or 25 μg/mL) in DMEM or RPMI 1640 medium for 24 h. Western blotting was performed to detect the expression of PRRSV N protein, FOS, and JUN proteins. (**B**,**C**) After treatment, PRRSV replication was assessed by TCID50 (**B**) and RT-PCR (**C**) 24 h later. (**D**,**E**) FOS and JUN mRNA expression levels were measured by qPCR 24 h post-treatment (**D**,**E**). (**F**) P38 and p-P38 protein levels were measured by Western blotting 24 h after treatment. *n* = 3. Bars represent means ± SEMs from three independent experiments. *** *p* < 0.001; ** *p* < 0.01; * *p* < 0.05. Note: ND (not detected).

**Figure 10 microorganisms-13-00450-f010:**
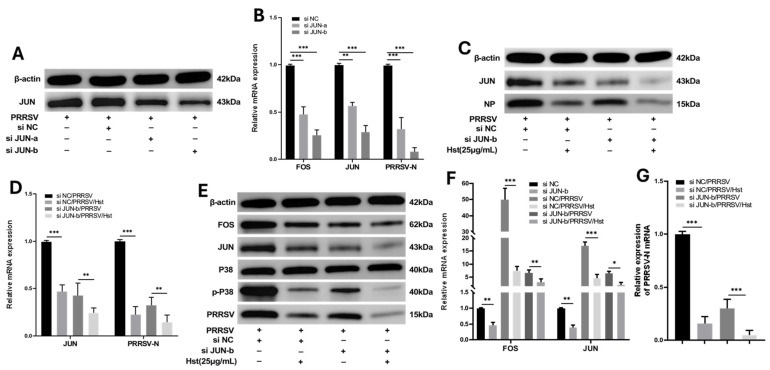
Knockdown of JUN gene inhibits PRRSV replication. (**A**,**B**) Marc-145 cells were transfected with 50 pmol of siNC or JUN siRNAs (siJUN-a and siJUN-b) using 5 µL of Lipofectamine RNAiMAX transfection reagent for 24 h, followed by PRRSV infection. Cells were harvested 36 h later for qRT-PCR (**B**) and Western blot (**A**) analysis. (**C**,**D**) Marc-145 cells were transfected with siJUN-b or siNC for 24 h, followed by PRRSV infection. After 2 h of treatment with 25 µg/mL Xn or DMEM, cells were harvested at 36 h for qRT-PCR (**C**) and Western blot (**D**) analysis. (**E**–**G**) Marc-145 cells were transfected with siJUN-b or siNC for 24 h, followed by PRRSV infection. After 2 h of treatment with 25 µg/mL Xn or DMEM, cells were harvested at 36 h. Western blotting was used to detect the protein expression levels of p-P38, P38, JUN, FOS, and PRRSV (**E**). qRT-PCR was used to measure the mRNA levels of PRRSV N protein (**F**). *n* = 3. Bars represent means ± SEMs from three independent experiments. *** *p* < 0.001; ** *p* < 0.01; * *p* < 0.05.

**Table 1 microorganisms-13-00450-t001:** Real-time PCR primer sequences.

Primer/Probe Name	Primer Sequence (5′→3′)	Product Size/bp
PRRSV-F	TTGCTAGGCCGCAAGTAC	183
PRRSV-R	ACGCCGGACGACAAATGC	
β-actin-F	TCCTGTGGCATCCATGAAACTA	284
β-actin-R	GACTCGTCATACTCCTGCTTGCT	
FOS-F	GAGCCCTTTGATGACTTCCTGTT	103
FOS-R	CTGCATAGAAGGACCCAGATAGG	
NR4A1-F	CTGGAGGTCATCCGCAAGTG	215
NR4A1-R	CTGTCAATCCAGTCGCCGAA	
DUSP1-F	AATGCTGGAGGAAGGGTGTTT	155
DUSP1-R	CTGAAGTTGGGGGAGATGATG	
JUN-F	AGATGGAAACGACCTTCTAC	110
JUN-R	CAGGGTCATGCTCTGTTTCA	

## Data Availability

All data are contained within the article.

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
