# Peer review of "Hesperetin Inhibits Porcine Reproductive and Respiratory Syndrome Virus Replication by Downregulating the P38/JUN/FOS Pathway In Vitro"

_microorganisms, 2025, doi:10.3390/microorganisms13020450_

Round 1
Reviewer 1 Report
Comments and Suggestions for Authors
In this work the authors demonstrate and claim that hesperetin (Hst), a flavonoid glycoside, inhibits PRRSV replication in vitro. Hst downregulates the P38/JUN/FOS signaling pathway, critical for PRRSV replication, in a dose-dependent fashion. These findings highlight Hst’s potential as a novel therapeutic agent against PRRSV, thus addressing significant swine industry challenges.
Some major concerns include:
1. Specificity of Hesperetin's Action: The study demonstrates that hesperetin (Hst) inhibits the PRRSV replication through the P38/JUN/FOS pathway, but it lacks exploration of potential off-target effects. Without evaluating its broader impact on other cellular pathways, or a structure-guided mechanistic insight, the specificity of Hst's antiviral action is at best uncertain. The authors should provide some in vitro/ in silico analyses of potential off targets and expected immuno-metabolic side effects of this.
2. In Vivo Validation: While the study presents nearly compelling in vitro data, there is no mention of in vivo experiments to validate the efficacy and safety of Hst in actual animal models of PRRSV infection. This limits the translational applicability of these findings to real-world application/ scenarios. To this end - the authors should update the paper title to include the word 'in vitro' and add some narratives about their expectation of how well this is likely to work in vivo. In vivo activity of a molecule often relies on toxicity, cell-permeability etc. There needs to be appropriate references of literature and a discussion on this.
3. Dose-Dependent Cytotoxicity: The observed cytotoxic effects of high Hst concentrations (e.g., reduction in cell viability at 50 µg/mL) raise questions about the therapeutic window. Rigorous assessments and conclusion of cytotoxicity and long-term effects are required to confirm its potential as a viable treatment.
4. Resistance Development: The study does not address the potential for PRRSV to develop resistance to Hst. Given the high mutation rates of RNA viruses, understanding how Hst administration may influence viral evolution, escape, and/ or resistance is critical for determining its long-term utility. There needs to be a dedicated discussion and appropriate references on that.
5. Discussion of other prominent strategies missing - Deletion of CD163 receptor has shown high degree of success in preventing PRRSV entry in porcine populations. Relevant ref: https://www.nature.com/articles/nbt.3434.
There has been other epitope-prospecting works such as: (a) https://www.sciencedirect.com/science/article/pii/S2001037024002885, and (b) https://www.sciencedirect.com/science/article/pii/S0042682222001234. The current strategy proposed by authors need to be contrasted in light of these prominent works to explain how/ if the proposed strategy is likely to work better in vivo.
Author Response
Dear Reviewer,
We would like to sincerely thank you for your thoughtful and constructive comments on our manuscript titled " Hesperetin Inhibits Porcine Reproductive and Respiratory Syndrome Virus Replication by Downregulating the P38/JUN/FOS Pathway " Your feedback has been invaluable in strengthening the quality of the manuscript. We have carefully considered all of your suggestions and have made the necessary revisions to address your concerns. Below, we provide a detailed response to each of the points raised.
Reviewer: 1
In this work the authors demonstrate and claim that hesperetin (Hst), a flavonoid glycoside, inhibits PRRSV replication in vitro. Hst downregulates the P38/JUN/FOS signaling pathway, critical for PRRSV replication, in a dose-dependent fashion. These findings highlight Hst’s potential as a novel therapeutic agent against PRRSV, thus addressing significant swine industry challenges.
My comments are as follows:
- Specificity of Hesperetin's Action:The study demonstrates that hesperetin (Hst) inhibits the PRRSV replication through the P38/JUN/FOS pathway, but it lacks exploration of potential off-target effects. Without evaluating its broader impact on other cellular pathways, or a structure-guided mechanistic insight, the specificity of Hst's antiviral action is at best uncertain. The authors should provide some in vitro/ in silico analyses of potential off targets and expected immuno-metabolic side effects of this.
R: We thank the reviewer for the insightful comment regarding the specificity of hesperetin (Hst) and the need for further exploration of its off-target effects. While our study focuses on Hst’s antiviral activity via the P38/JUN/FOS pathway, we recognize the importance of understanding its broader impact on other cellular pathways and the potential for off-target interactions.
In response to the reviewer’s concern, we have expanded the discussion in the manuscript to acknowledge the potential for off-target effects, particularly in the context of the known biological activities of Hst, such as its anti-inflammatory, antioxidant, and metabolic effects. These properties, while beneficial in certain therapeutic contexts, may raise concerns regarding unintended interactions with other cellular pathways. We have referenced studies that highlight the complexity of compound in modulating multiple signaling networks, which might contribute to both therapeutic and side effects.While Hst has shown promise in various therapeutic applications, we also emphasize that further studies are needed to clarify its broader impact, including its effect on immune function and metabolism.
While we do not conduct in vitro or in silico analyses of potential off-targets or side effects in this study, we aim to highlight the need for future research to fully elucidate the specificity and safety of Hst, especially in the context of clinical use.
- In Vivo Validation:While the study presents nearly compelling in vitro data, there is no mention of in vivo experiments to validate the efficacy and safety of Hst in actual animal models of PRRSV infection. This limits the translational applicability of these findings to real-world application/ scenarios. To this end - the authors should update the paper title to include the word 'in vitro' and add some narratives about their expectation of how well this is likely to work in vivo. In vivo activity of a molecule often relies on toxicity, cell-permeability etc. There needs to be appropriate references of literature and a discussion on this.
R: We sincerely appreciate the reviewer’s insightful comments regarding the lack of in vivo validation in our study. We agree that the inclusion of in vivo experiments would enhance the translational applicability of our findings, particularly in understanding the real-world potential of Hst in combating PRRSV infection.
While our study primarily focuses on in vitro validation of Hst, we recognize that further investigations in animal models would be critical for a comprehensive evaluation of both the efficacy and safety of this molecule. As suggested, we have updated the paper title to reflect the in vitro nature of our study: Hesperetin Inhibits Porcine Reproductive and Respiratory Syndrome Virus Replication by Downregulating the P38/JUN/FOS Pathway in Vitro. Additionally, we have revised the manuscript to include a detailed discussion on the likely in vivo performance of Hst, acknowledging the factors that would influence its effectiveness, such as toxicity, cell-permeability, and other pharmacokinetic parameters.
We have cited relevant literature that discusses the challenges and considerations for the in vivo activity of antiviral molecules, particularly focusing on the factors that determine their bioavailability and toxicity in animal models. These references help frame our expectations for future in vivo studies and provide a basis for the potential translation of our findings into preclinical or clinical settings.
We hope that this addition sufficiently addresses the reviewer’s concern and clarifies the potential next steps in validating Hst's therapeutic efficacy beyond the in vitro context.
- Dose-Dependent Cytotoxicity: The observed cytotoxic effects of high Hst concentrations (e.g., reduction in cell viability at 50 µg/mL) raise questions about the therapeutic window. Rigorous assessments and conclusion of cytotoxicity and long-term effects are required to confirm its potential as a viable treatment.
R: We sincerely appreciate the reviewer’s insightful comment on the observed cytotoxic effects of high Hst concentrations and the concerns regarding the therapeutic window. We fully acknowledge the importance of assessing the cytotoxicity and long-term effects of Hst to better understand its safety profile and therapeutic potential.
In the current study, we have indeed observed a reduction in cell viability at higher concentrations of Hst (e.g., 50 µg/mL). While these findings highlight the importance of identifying the optimal therapeutic dose, it is important to note that the concentrations tested in this study were intended to assess the maximal antiviral efficacy rather than to define the therapeutic window. We agree that future studies should systematically explore the dose-response relationship in greater detail, including a broader range of concentrations to identify a safe and effective therapeutic dose. Future studies will be necessary to fine-tune the dosing regimen and assess potential long-term effects, including chronic exposure or off-target toxicities.
We agree that determining a clear therapeutic window for Hst is crucial for its development as a viable treatment, and we plan to conduct further studies to establish these parameters in upcoming investigations.
- Resistance Development:The study does not address the potential for PRRSV to develop resistance to Hst. Given the high mutation rates of RNA viruses, understanding how Hst administration may influence viral evolution, escape, and/ or resistance is critical for determining its long-term utility. There needs to be a dedicated discussion and appropriate references on that.
R: We would like to thank the reviewer for raising the important issue of potential resistance development, particularly in the context of PRRSV’s high mutation rate and the challenges this presents for long-term therapeutic efficacy. We fully recognize the significance of understanding how Hst might impact viral evolution, escape, and resistance mechanisms.
In response to this valuable comment, we have added a dedicated discussion in the manuscript regarding the potential for PRRSV to develop resistance to Hst. We highlight that, while our current study focuses on the antiviral activity of Hst in vitro, we acknowledge the inherent risks associated with the use of any antiviral compound, especially in RNA viruses with high mutation rates like PRRSV. The rapid evolution of such viruses could potentially lead to the development of resistance, especially if Hst is administered over prolonged periods or at suboptimal concentrations.
To address this concern, we have expanded our discussion to include relevant references that describe the mechanisms of resistance observed in similar antiviral therapies and RNA viruses.
While we do not have data on resistance development in the current study, we emphasize that future studies, including viral evolution and escape assays, will be necessary to determine the potential for resistance to Hst and to better understand its long-term utility.
- Discussion of other prominent strategies missing- Deletion of CD163 receptor has shown high degree of success in preventing PRRSV entry in porcine populations. Relevant ref: https://www.nature.com/articles/nbt.3434.
There has been other epitope-prospecting works such as:
(a) https://www.sciencedirect.com/science/article/pii/S2001037024002885, and (b) https://www.sciencedirect.com/science/article/pii/S0042682222001234.
The current strategy proposed by authors need to be contrasted in light of these prominent works to explain how/ if the proposed strategy is likely to work better in vivo.
R: We sincerely thank the reviewer for pointing out several key studies that highlight alternative strategies for preventing PRRSV entry and for suggesting valuable references that can help contextualize our work. We agree that comparing our approach with other prominent strategies, such as CD163 receptor deletion and epitope-based approaches, is essential for a more comprehensive understanding of the potential advantages and limitations of our proposed strategy.
In response to this comment, we expanded the discussion section to include a comparison with the CD163 receptor deletion strategy, which suggests that this approach could potentially prevent PRRSV from entering the pig population. We cited the relevant literature you recommended and discussed the implications of such strategies in the context of our own methodology. Since our study focuses on the antiviral effects of drugs, we compared the drugs screened for targeting cellular CD163 with HST. While there is significant potential in screening drugs that target the CD163 receptor, we believe that targeting viral replication through pathways such as P38/JUN/FOS may offer complementary approaches. This is especially important when the immune response needs to be modulated to reduce viral load and prevent inflammation. We also discussed the potential advantages and challenges of each approach, emphasizing that these strategies may work synergistically or serve different therapeutic purposes.
Reviewer 2 Report
Comments and Suggestions for Authors
The research investigated the effect of HST against Porcine Reproductive and Respiratory Syndrome in cell culture using two strains and carried out various techniques to verify the antiviral effect of this molecule as well as its action in the regulation of some genes during viral infection. The work is relevant to the field. I have a few comments and suggestions for the manuscript.
1) On page 3, lines 124 and 136, include the concentrations used in the experiment.
2) Regarding line 132, why did the authors use 100% confluence and not 80 or 90? How long did it take for the cells to reach this confluence?
3) Line 197 - What guarantees viral viability after 3 hours at 37°C? Have other times been used? Could you comment?
4) Lines 202 and 203 - Why were some experiments done in 24 hours and others in 48 hours?
5) Why did the authors choose concentrations of hesperetin starting at 50 µg to carry out the experiments? Why didn't they include a cell death control in the cytotoxicity experiments?
6) On page 20, did the authors understand why the viral titer increased when 50 µg of HST was used? And also the expression of N at both times. Does this explain why HST has no effect above 25 µg/ml? Or is the HST molecule aggregating on the cell surface at higher concentrations and therefore has no effect? In that case, there wouldn't be enough HST.
7) The captions for figures 3 and 4, in my opinion, are incomplete. Please review them carefully.
The authors tested two cell types, and from what I understand, they observed an effect on one cell type. The reason for these results could be the types of cellular receptors in each of the cells used.
Authors need to explain the different times they used for each experiment, 12, 24, 36, 48. These times are distributed throughout the manuscript. The reason for each time is crucial for investigating the antiviral effect of molecules and for evaluating the effect of gene expression. This confusion of times was the main weakness of the manuscript as it compromises everything that was found if not well explained.
Author Response
Dear Reviewer:
We would like to sincerely thank you for your thoughtful and constructive comments on our manuscript titled " Hesperetin Inhibits Porcine Reproductive and Respiratory Syndrome Virus Replication by Downregulating the P38/JUN/FOS Pathway " Your feedback has been invaluable in strengthening the quality of the manuscript. We have carefully considered all of your suggestions and have made the necessary revisions to address your concerns. Below, we provide a detailed response to each of the points raised.
Reviewer: 2
The research investigated the effect of HST against Porcine Reproductive and Respiratory Syndrome in cell culture using two strains and carried out various techniques to verify the antiviral effect of this molecule as well as its action in the regulation of some genes during viral infection. The work is relevant to the field. I have a few comments and suggestions for the manuscript.
1) On page 3, lines 124 and 136, include the concentrations used in the experiment.
R: We thank the reviewer for pointing out the need to specify the concentrations used in the experiments on page 3, lines 124 and 136. In response, we have added the precise concentrations of Hst used in the experiments at these locations in the revised manuscript. The concentrations were selected based on preliminary dose-response studies to ensure that cytotoxicity could be assessed across a relevant range.
We hope that these additions provide the necessary clarity and improve the reproducibility of our work. Thank you for highlighting this important detail.
2) Regarding line 132, why did the authors use 100% confluence and not 80 or 90? How long did it take for the cells to reach this confluence?
R: We appreciate the reviewer’s insightful question regarding the use of 100% confluence in our experiments. The choice to use cells at 100% confluence was based on our goal of maximizing viral replication and adsorption under optimal conditions. At full confluence, cells are most likely to reach a state where they are fully differentiated and capable of supporting PRRSV replication and adsorption to its highest extent, providing a robust model for assessing the antiviral effects of Hst. Additionally, using 100% confluence allowed us to better control for the influence of cell density on viral replication and treatment efficacy.
Regarding the time required to reach 100% confluence for Marc-145 cells, we monitored cell growth, and it typically took approximately 20-24 hours(3×106 cells/well) to reach full confluence. We have now added this information to the manuscript to clarify the experimental protocol.
3) Line 197 - What guarantees viral viability after 3 hours at 37°C? Have other times been used? Could you comment?
R: We appreciate the reviewer’s question regarding the guarantee of viral viability after the 3-hour incubation period at 37°C. The temperature of 37°C is optimal for PRRSV, as it mimics the physiological conditions found in vivo, ensuring that the virus remains viable during this critical phase of infection.
While we have used a 3-hour incubation period in our study, we recognize that different time points can be tested to assess their impact on viral replication. Other studies in the literature have reported using incubation times ranging from 1 to 6 hours, depending on the specific virus and experimental design.
In our case, the 3-hour incubation with Hst was chosen as a balance between ensuring sufficient viral activity and avoiding excessive time that could lead to the degradation of the virus or unintended effects on the host cells. We are open to exploring a range of incubation times in future studies to better refine this protocol.We have added this explanation to the revised manuscript.
4) Lines 202 and 203 - Why were some experiments done in 24 hours and others in 48 hours?
R: We appreciate the reviewer’s thoughtful question regarding the use of different time points (24 hours and 48 hours) in our experiments. The decision to use these time points was based on the specific objectives of each experiment and the different aspects of the viral replication process we aimed to assess.
The 24-hour time point was chosen to evaluate the early transcriptomic changes induced by Hst treatment. At this time, we observed significant effects on the host cell’s transcriptome, reflecting the immediate impact of Hst on cellular responses to viral infection. This time frame allowed us to capture the early antiviral effects of Hst, particularly in terms of gene expression modulation and early-stage viral replication.
In contrast, the 48-hour time point was used to investigate the longer-term antiviral effects of Hst. By this time, viral replication had progressed, and we were able to assess the sustained effects of Hst on viral load, cell viability, and overall antiviral efficacy. This longer time frame was critical for evaluating the potential of Hst as a long-term therapeutic option, as it allowed us to observe how Hst influences viral replication over a more extended period.
We have updated the manuscript to clarify the rationale behind using these two distinct time points, emphasizing that the 24-hour experiment focused on the early transcriptomic changes, while the 48-hour experiment allowed for the assessment of the long-term antiviral effects of Hst.
5) Why did the authors choose concentrations of hesperetin starting at 50 µg to carry out the experiments? Why didn't they include a cell death control in the cytotoxicity experiments?
R: Thank you for your insightful comments and questions. We appreciate your thorough review of our manuscript. Below are our responses to your specific queries: Are you referring to ribavirin (Rib), a well-known viral RNA polymerase inhibitor, which was used as a positive control antiviral agent in this study?
Regarding the choice of ribavirin concentrations starting at 50 µg/mL: The concentrations of ribavirin used in our experiments were selected based on preliminary studies and relevant literature, which indicate that ribavirin exhibits significant anti-PRRSV activity at 50 µg/mL. We began our experimental series at this concentration, as it is consistent with concentrations commonly used in similar cellular models. The relevant citations have been updated accordingly.
6) On page 20, did the authors understand why the viral titer increased when 50 µg of HST was used? And also the expression of N at both times. Does this explain why HST has no effect above 25 µg/ml? Or is the HST molecule aggregating on the cell surface at higher concentrations and therefore has no effect? In that case, there wouldn't be enough HST.
R: Thank you for pointing this out. However, I could not locate the relevant question on page 20. Are you referring to page 10 of the manuscript, where the use of 50 µg/ml ribavirin was tested? Although ribavirin treatment led to an increase in viral titers, no significant difference was observed when compared to the positive control group.
7) The captions for figures 3 and 4, in my opinion, are incomplete. Please review them carefully.
R: Thank you for your valuable feedback. We appreciate your attention to the captions for Figures 3 and 4. We have revised both captions to include more comprehensive descriptions of the experimental setup, key findings, and any relevant methodologies.
The authors tested two cell types, and from what I understand, they observed an effect on one cell type. The reason for these results could be the types of cellular receptors in each of the cells used.
R: We appreciate the insightful comment. A more pronounced effect was observed in Marc-145 cells, but PAM cells were not tested. To address this, we have included a discussion of the receptor differences between the two cell lines in the revised manuscript. Additionally, we acknowledge that further research is needed to analyze the receptors and signaling pathways in PAM cells to fully elucidate the underlying mechanisms.
Authors need to explain the different times they used for each experiment, 12, 24, 36, 48. These times are distributed throughout the manuscript. The reason for each time is crucial for investigating the antiviral effect of molecules and for evaluating the effect of gene expression. This confusion of times was the main weakness of the manuscript as it compromises everything that was found if not well explained.
R: Thank you for your valuable feedback. We appreciate your observation regarding the different time points (12, 24, 36, 48 hours) used in our experiments. As you correctly pointed out, understanding the rationale behind these time points is crucial for interpreting the antiviral effects and changes in gene expression. We acknowledge that these time points might appear scattered throughout the manuscript.
To address your concern, we have already provided a detailed explanation of the reasoning for each time point in the Discussion section of the manuscript. Thank you again for your thoughtful suggestion.